

# Icelandic Snow Cover Characteristics derived from a gap-filled MODIS Daily Snow Cover Product

Andri Gunnarsson[1,2], Sigurður M. Garðarsson[1], and Óli G.B. Sveinsson[2]

[1]University of Iceland, Civil and Environmental Engineering, Hjardarhagi 2-6, IS-107 Reykjavik, Iceland
[2]Landsvirkjun, Department of Research and Development, Reykjavík, IS-107, Iceland

**Correspondence:** Andri Gunnarsson (andrigun@lv.is)

**Abstract.** This study presents a spatio-temporal continuous data set for snow cover in Iceland based on the Moderate Resolution Imaging Spectroradiometer (Modis) from 2000 - 2018. Cloud cover and polar darkness are the main limiting factors for data availability of remotely sensed optical data at higher latitudes. In Iceland the average cloud cover is 75 % with some spatial variations and polar darkness reduces data availability from the Modis sensor from late November until mid January. In this study Modis snow cover data were validated over Iceland with comparison to manned in-situ observations, Landsat 7/8 and Sentinel 2 data. Overall a good agreement was found between in-situ observed snow cover with an average agreement of 0.925. Agreement of Landsat 7,8 and Sentinel 2 was found to be acceptable with $R^2$ values 0.96, 0.92 and 0.95, respectively, and in agreement with other studies. By applying daily data merging from Terra and Aqua and temporal aggregation of 7 days, unclassified pixels were reduced from 75 % to 14 %. The remaining unclassified pixels after daily merging and temporal aggregation were removed with classification learners trained with classified data, pixel location, aspect and elevation. Various snow cover characteristic metrics were derived for each pixel such as snow cover duration, first and last snow free date, deviation and dynamics of snow cover and trends during the study period. On average the first snow free date in Iceland is June 27 with a standard deviation of 19.9 days. For the study period a trend of increasing snow cover duration was observed for all months except October and November. However, statistical testing of the trends indicated that there was only a significant trend in June.

*Copyright statement.* TEXT

## 1 Introduction

On a global scale snow cover has a strong interaction with the cryosphere and ocean systems and therefore the climate system of the Earth. The two main effects of snow on the cryosphere are its control on the reflection of radiation, reaching the surface of Earth and balancing its radiation budget (Barry, 2002; Warren, 1982) and low thermal conductivity which is dominating for the growing season length of vegetation and plants (Keller et al., 2005). Snow albedo dominates the control of its irradiance feedback which depends on various factors such as snow depth, snow cover extent, vegetation and cloud cover (Fernandes et al.,



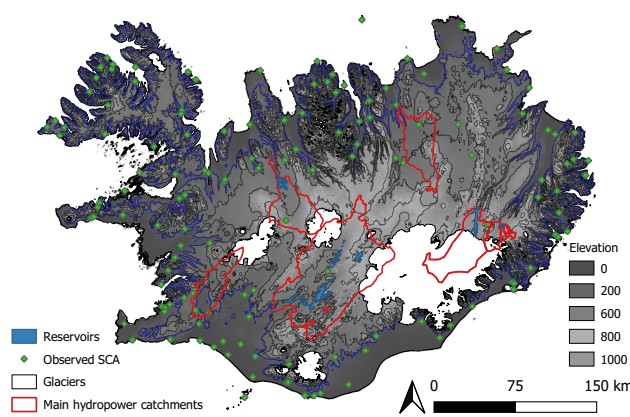

**Figure 1.** Overview of Iceland. Red outlines show main catchment boundaries for large hydropower diversions, reservoirs and power plants. Manned sites for observed snow cover are shown in green points. Contours are shown for 200 m elevation band. A solid blue contour represents 200 m elevation.

2009; Qu and Hall, 2007). In the Northern Hemisphere the spring snow cover extent has decreased significantly, influencing the dynamics of spring melt intensity and timing in recent years (Adam et al., 2008; Barnett et al., 2005). Future projections with warming trends predict less precipitation to fall as snow and snow melt to occur earlier in spring, affecting runoff and water resources downstream (Vaughan et al., 2013; IPCC, 2013). On regional scales, seasonal snow is a vital part of water budgets in mountain and highland catchments where precipitation falls as snow during winter (Raleigh et al., 2013). Seasonal snow spring melt is also important for many applications such as irrigation for downstream agriculture areas, drinking water supply, availability of water for hydropower energy production and in some regions critical for tourism, in particular ski resorts and winter tourism (Fischer et al., 2011; Kiparsky et al., 2014; Jóhannesson et al., 2007; Wagner et al., 2016).

Iceland is an island with an area of 103.100 km$^2$ located in the North Atlantic Ocean, close to the Arctic Circle (between 63° N and 66° N). The central highlands correspond to 40 % of the island with an average altitude of 550 m a.s.l. and only a quarter of the island lies below 200 m a.s.l. (Fig 1,2). About 50 % of Icelands land area is classified as open spaces and bare soils with sparse vegetation and 37 % as semi-natural vegetation, these two types include most of the central highlands. Less than 1 % is forested and in general low shrub, wetland and heathland are the main types of vegetation (Einarsson et al., 2005; Traustason and Snorrason, 2008). Precipitation climatology has been characterized by a precipitation reduction with higher latitudes controlled by the orographic generation of precipitation in mountainous regions corresponding to the dominating SE to SW wind direction (Crochet et al., 2007; Björnsson et al., 2018). Area average precipitation is 1.7 mH2O with the highest values at glacier peaks in the south (up to 10 mH2O). During winter heavy snowfall is frequently induced by cyclones crossing the North Atlantic, where air and water masses of tropical and arctic origins meet (Einarsson, 1984; Ólafsson et al., 2007). In the highlands this leads to the formation of a seasonal snow pack and the sustainment of higher altitudes glaciers. At present, about 11 % of the country is covered by glaciers (Björnsson and Pálsson, 2008) ((Fig 1,2)). During summer average temper-





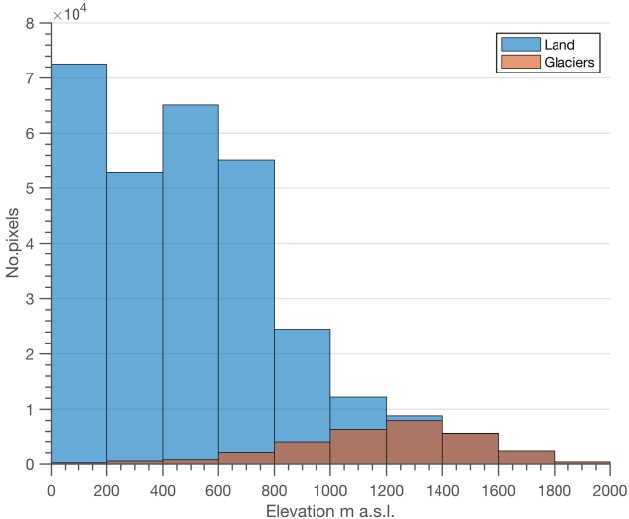

**Figure 2.** Elevation distribution for Iceland for both land and glaciers. Glaciers cover about 11 % of Iceland.

ature at lower elevations (< 400 m a.s.l.) range from 8°C - 10°C with a country-wide average of 7°C. In winter the average temperature is 0°C to -3°C at lower elevations and about -5°C for the whole island (Björnsson, 2003; Henriksen, 2003). From the seasonal snow cover classification system proposed by Sturm et al. (1995, 2010) Icelandic snow pack generally classifies as a combination of taiga, tundra and maritime types with overall shallow snowpack in depth with high density, frequent melt

and wind blown features (Jóhannesson and Sigurðsson, 2014).

In Iceland runoff from snow melt is critical for hydropower production and reservoir storage as the energy system is strongly dependent on snow and glacier melt. Over 13 % of the highland area in Iceland is developed for hydropower generation which provides over 72 % of the total average energy produced in Iceland (Hjaltason et al., 2018). A system of reservoirs and di-

versions store melt water during the spring freshet which generally consists of a seasonal snow melt period (April - June), a glacier melt period (June - September) and precipitation in the fall (August - October). During winter reservoir storage provides regulation of water resources for energy production. The isolation and high natural climate variability poses a risk to the energy security of the power system as drought conditions and low flow periods are usually not foreseen. In the longer term inflow to the energy systems is projected to increase due to climate warming and associated increase in glacier melt (Jóhannesson et al.,

2007). Flow dynamics, i.e. timing and magnitude for seasonal snow will also change, posing a challenge for operational control of energy infrastructure and climate change adaptation both for current energy projects but as well for future development (Björnsson and Thorsteinsson, 2012; Sveinsson, 2016).

Space borne sensors operating in the visible and near-infrared range of electromagnetic spectrum have proven to be useful

to effectively map snow cover for large areas since the early 1980s (Baumgartner et al., 1987; Dozier and Marks, 1987). Snow



cover extent maps at various resolutions have been derived by the National Oceanic and Atmospheric Administration (NOAA) since 1966 (Dewey, 1987; Matson, 1991; Robinson et al., 1993). Since 2000 the MODIS sensor (Moderate Resolution Imaging Spectroradiometer) provides daily global coverage of snow cover in cloud free areas at a spatial resolution ranging from 250 m to 1000 m. The sensor is carried on two sun-synchronous, near-polar circular orbit satellites, Terra (descending node at

approximately 10:30 A.M. local time) and Aqua (ascending node at approximately 1:30 P.M. local time). Terra was launched on December 18,1999 and has had data available since September 2000 while Aqua was launched in May, 2002. The sensor has 36 spectral bands that are used for various cryosphere, land, ocean and atmospheric scientific data sets and applications.

A range of snow cover products has been developed from the Aqua and Terra satellites carrying the MODIS sensor dating back to the early 2000s (Hall et al., 2002). The MODIS daily snow cover products (MOD10A1 from Terra and MYD10A1

from Aqua) are a standard for snow cover monitoring at medium resolution since mid-year 2000 and are commonly used to analyze and monitor snow cover development in snow-dominated catchments and their near real time availability makes them desirable for real time applications such as for short term forecasting and validation of runoff. The discriminating of snow and land is based on the Normalized Difference Snow Index (NDSI) which utilizes the spectral signature of snow being highly reflective in the visible spectral range (VIS) and has very low reflectance in the shortwave infrared spectral ranges (IR). In

Modis Terra bands 4 (VIS / 0.545–0.565 $\mu$m) and 6 (IR / (1.628–1652 $\mu$m)) are used for the NDSI calculations while the Modis Aqua product relies on bands 4 and 7, as band 6 is non-functional (Salomonson and Appel, 2006).

Modis snow cover products have been widely tested and validate for various land covers, topographic regions, and climates with a typical average absolute accuracy of 93 % (Hall and Riggs, 2007; Huang et al., 2011; Klein and Barnett, 2003; Parajka

and Blöschl, 2006). One of the main drawbacks of Modis snow cover products, as well as other products that rely on optical satellite sensors, is the reliance on cloud free conditions to produce snow cover maps. Various methods have been tested to provide gap-filled products of optical remote sensing products including snow cover. Gao et al. (2010) used the MODIS high spatial resolution and cloud penetrating ability of AMSR-E to reduce gaps in snow cover maps while (Gascoin et al., 2015) applied a classification tree to gaps after merging daily Aqua and Terra snow cover tiles together and applying a temporal ag-

gregating filter. Data from higher spatial resolution satellite platforms are available from high resolution visible/near-infrared sensors such as from the USGS Landsat program ( 30m) and ESA Sentinel 2 ( 20m) program but at a lower temporal resolution, often making them less attractive for operational observations of snow cover.

The main objective of this study was to create a gap-filled snow cover product for Iceland and extract snow cover character-

istics for the period from 2000 to 2018. The first objective was a thorough validation of Modis sensor derived snow-covered maps over Iceland to validate the quality of the product and assess its limitations. Validation was an important and necessary step due to the annual and seasonal variability in climate, high average cloud cover and polar darkness during winter. The second objective of the study was to reduce the gaps to provide a spatio-temporal continuous product. By merging of data and temporal aggregation methods data gaps are reduced and finally eliminated by using classification learners trained on topog-





raphy and location of pixels. Based on the gap-filled dataset snow cover characteristics on a regional scale over Iceland were derived showing relations to elevation, aspect and general trends in snow cover extent and duration.

## 2    Data

### 2.1    Observational in-situ data

In Iceland in-situ snow cover and depth observations are sparse, especially in the highlands. Few sites have automatic observations of properties of snow until recently. The Icelandic Meteorological Office (IMO) operates a network of synoptic meteorological observations including daily manned observation of snow cover at 9 a.m. Figure 1 shows the location of these sites (green points) and how few of them are in or near the central highland area. Data were obtained from the IMO for the time period from 1.2.2000 to 31.12.2017 spanning in a total of 152 sites and 585.880 observations. The dataset consists of daily observations with a site number, date and snow cover classifications as well as a metadata file with site number, site name and site location and elevation (Veðurstofa Íslands, 2017).

### 2.2    MODIS Snow cover data

MOD10A1 (Terra) and MYD10A1 (Aqua) Version 6 were obtained from the National Snow and Ice Data Center (NSIDC) (Hall and Riggs, 2016a), (Hall and Riggs, 2016b) for the period from 23.02.2000 to 31.06.2018 which corresponds to 6702 dates where 6640 (99 %) MOD10A1 granules were available and 5829 (87 %) MYD10A1 dates were available. For MOD10A1 62 dates were missing and 12 for MYD10A1 from NSIDC excluding data missing due to polar darkness. Polar darkness limits the data availability during winter from MODIS in Iceland from  20th November until January 26 (63 days) each year reducing the dataset during winter (Dietz et al., 2012). During Polar Darkness M*D10A1 snow product pixels are classfied as Night when the solar zenith angle is larger or equal to 85°. Every granule from tile h17v02 was used in this project as it covers all the central highlands in Iceland and leaves out only a small portion of the west Snæfellsnes Peninsula and the Westfjords.

### 2.3    Landsat 7/8 and Sentinel 2 data

Data acquired by Landsat 7 Thematic Mapper (TM), Landsat 7 Enhanced Thematic Mapper Plus (ETM+), Landsat 8 Thematic Mapper (TM) and Landsat 8 Thermal Infrared Sensor (TIRS) were used. The data were downloaded from the United States Geological Survey (USGS) (https://earthexplorer.usgs.gov/) using bulk download utilities. Landsat scenes that cover Iceland are numbered from 224-13 to 216-13, 224-14 to 215-14, 223-15 to 215-15 and 219-16 to 216-16 in the Worldwide Reference System 2 (WRS2), a total of 32 Landsat footprints(USGS, 2018). In total 264 Landsat 7 scenes were available from 12.04.2000 to 01.04.2015 and 124 scenes from Landsat 8 from 26.04.2013 to 22.06.2018 where land cloud cover was equal or less than 20 % and solar zenith angle not too large for processing the scene.



Data acquired by ESA Sentinel 2A and B Multispectral instrument (MSI) sensor were also used. The data were downloaded from the European Space Agency (ESA) datahub (https://scihub.copernicus.eu/dhus). In total 1090 Sentinel 2A/B scenes were acquired from 21 tracks covering Iceland. Only images where land cloud cover was equal or less than 20 % were used. Images acquired in December and January each year have been left out due to polar darkness of Modis data for all satellite products.

Both Landsat and Sentinel products are in UTM/WGS84 projection.

## 2.4    Geospatial data

Digital elevation models and masking data for water bodies and glaciers were obtained from the National Land Survey of Iceland. The original DEM is a raster with a 10 m spatial resolution which is resampled to match the grid of the MODIS pixels using nearest neighbour sampling. From the resampled 500 m DEM the aspect data are calculated.

# 3    Methods

## 3.1    Landsat and Sentinel 2A/B processing

Landsat 7 and 8 data were retrieved as L1TP surface reflectance products. These products have a terrain correction and are radiometricly calibrated. U.S. Geological Survey (USGS) uses the Landsat Ecosystem Disturbance Adaptive Processing System (LEDAPS) for Landsat 7 Surface Reflectance generation while Landsat 8 is processed with Landsat Surface Reflectance Code

system (LaSRC) (USGS, 2018). Outputs from these systems include for each date tile a pixel quality map where classifications of clouds, shadows, land, water and snow is presented.

Sentinel 2 data were retrieved as a L1C orthoimage product. Images are in Top-of-atmosphere reflectances in cartographic geometry and have undergone geometric transfomation and radiometric interpolation with a constant GSD (Ground Sampling Distance) (Delwart, 2015). Sentinel 2 L1C data were processed using the Sen2Corr application from ESA to produce Level 2A

data (Louis et al., 2016; Müller-Wilm et al., 2013). Level 2A Sen2Corr output data were atmosphericly corrected bottom of the atmosphere (BOA) product and has a scene classification map (SCL) on a pixel basis discriminating the surface to 11 categories including no data, various types of clouds, water and snow or ice among other categories. The data were processed at 20 m spatial resolution and then resampled to the Modis data grid at 500 m spatial resolution. Classification from the classification map in the L2A product were used for further analysis. Snow and ice were classified with the NDSI method (Uwe Müller-Wilm,

2018; Salomonson and Appel, 2004).

## 3.2    in-situ data processing

Manned observations of snow cover from the IMO are reported daily at 9 a.m. Observations are made both at the local site where the instruments are located but as well in mountains where applicable, these are reported as local snow cover (SNC) and snow cover in mountains (SNCM). For each observation the local snow cover is reported as snow free (Code 0), patchy snow

cover, (Code 2) and fully snow covered (Code 4) (Veðurstofa Íslands, 2008). In accordance to the observational procedure





of local snow cover the area observed was within 1 km of the observer and had not more than a 50 m elevation difference. We only used the local snow cover (SNC) for our analysis and omitted patchy snow cover classification from our comparison but no further adjustments were made to the dataset. In total 213.011 observations matches are found, i.e. where a manned observations was available and cloud free pixel from MCDAT.

## 3.3 MODIS snow cover product processing

From the MOD10A1 and MYD10A1 daily data tiles we extracted the MOD Grid Snow 500 m grid and the variable NDSI Snow Cover was used for further analysis of snow cover. It is based on the MOD10-L2 algorithm which selects the best observation of the day to write to the daily dataset. The variable NDSI snow cover ranges from 0-100 but in addition various other classifications are provided with the tile. As a preprocessing step data was reclassified to a) Snow, b) no snow (Land) and c) no data (clouds, missing data, no decision, saturated detector). As the spatial extent of the tile is  1200 km x 1200 km (data dimension 2400 x 2400) values that are beyond the Icelandic coast were masked out including values only on land. A processing pipeline of Modis snow data was adopted from Gascoin et al. (2015) and Parajka and Blöschl (2008) with modifications. The main steps are

– 1. Daily tile merging: Daily tiles from Aqua and Terra are merged to a single dataset to improve daily coverage with data. Data from Terra has priority over data from Aqua as previous studies have found data from Terra to be of higher accuracy (Gascoin et al., 2015). For the first two years only Terra was in orbit so for the period from 23.02.2000 to 04.05.2002 is only based on Terra. The output dataset used for further processing is named MCDAT.

– 2. Temporal aggregation: For the remaining unclassified pixels in the daily merged data tiles (MCDAT) we apply temporal aggregation to further reduce unclassified pixels due to clouds in the data. Each MCDAT tile from step 1 is given a center date as the date of acquisition (t = 0) and a temporal aggregation range selected. The temporal aggregation range is set as number of days backwards and forwards each center date data is allowed to search for classified pixel data which are missing in the original MCDAT center date data tile. Priority is given to data closest to the center date data (newest data relative to the center date) and from the forward date if both backward and forward dates have data. We select a temporal aggregation range as 3 days backward/forward (t = +/- 3 days), i.e. in total 7 days can contribute data to the temporal aggregation product. The output dataset used for further processing is named MMCDDATA7D

– 3. Gap filling with classifiers: After the first two processing steps the remaining gaps are classified as snow or no snow with classification learners. For each dataset the unclassified pixels are reclassified with four predicting variables, location (easting, northing), elevation (Z) and aspect. The final output dataset used for further processing is named MCD7GFD.

## 3.4 Classification learners

To classify the data we need to select a classification method. In general terms a model is trained with the training dataset and the trained fit applied to the data that need classification. Within Matlab Classification Toolbox (Matlab, 2017) there are many methods and algorithms available and no clear selection criteria are evident.



In general snowfall in a region and formation of a snowpack are dependent on several climate and geographic factors such as latitude, longitude, elevation, distance from moist sources (ocean and lakes) and regional air mass circulation (DeWalle and Rango, 2008; Gray and Male, 2004). To classify the remaining unclassified pixels information about location (Latitude, Longitude), elevation and aspect to account for earlier melting of south facing slopes are derived to apply for a gap filling

algorithm. To test different classification methods a simple workflow was applied where the pre-processed Modis dataset (after temporal aggregation) 2400 by 2400 pixels for the tile covering (hv017v14) Iceland was masked with the coastline of Iceland selecting only pixels that fell on land reducing the data size from 5.75 million pixels to 472 thousand pixels. Next all pixels were categorized depending on whether they had data or not (snow/no snow/cloud). All cloud covered pixels are arranged to a classify data set and pixels with valid snow cover data were arranged to a training dataset. Information on location (X,Y),

elevation (Z) and aspect (A) data for each pixel was derived to train the classifiers. Finally the classifier was applied to the training dataset and the classified dataset reclassified as snow/no snow with the trained classification dataset. To assess the accuracy of the classification method 25 % of the classified dataset prior to classification was withheld for cross correlation. Glaciers are set to a fixed snow cover and water bodies were masked out.

**4   Results and discussion**

### 4.1   Validation results

#### 4.1.1   Comparison with observed snow cover

Overall a good agreement was found between in-situ observed snow cover and Modis daily combined snow cover (MCDAT). Figure 3 shows the average agreement for each of the 152 sites investigated compared to the Modis product for the whole period

where the circle size shows number of observations for each site. Out of the 585.800 observations in the database 213.011 matches were found when data were available from MCDAT daily product and manned observations. The average agreement between observed snow and Modis was 0.925. Table 1 shows a confusion matrix for the agreement between manually observed snow and NDSI snow cover from MCDAT. Observations and MCDAT agreed in 96.9 % of the time when there was no snow on the ground according to the manual observations and in 88.6 % of the time when snow was present. The poorest agreement

was for sites located in the bottom of fjords and sounds where snow was observed during the manual observation but was not present at the 10:30/01:30 UTC Terra/Aqua overpass.

#### 4.1.2   Comparison to Landsat/Sentinel data

Table 2 shows a confusion matrix classification results from pixel based comparison of snow cover derived from the combined daily Aqua and Terra product from Modis and snow cover derived from Landsat 7, 8 and Sentinel 2. For Landsat 7 8.21 million

pixels were compared, 8.6 million for Landsat 8 and 3.77 million for Sentinel 2. For Landsat 7 86.9 % of snow covered pixels were correctly classified in MCDAT product while 93.2 % of snow free pixels were correctly classified. For Landsat 8 83.8 %



**Table 1.** Confusion matrix for observations of snow compared to Modis Aqua and Terra daily snow product combined.

|  |  | MCDAT | |
| --- | --- | --- | --- |
|  |  | No snow | Snow |
| Obs. | No snow | 96.9% | 3.1% |
|  | Snow | 11.4% | 88.6% |

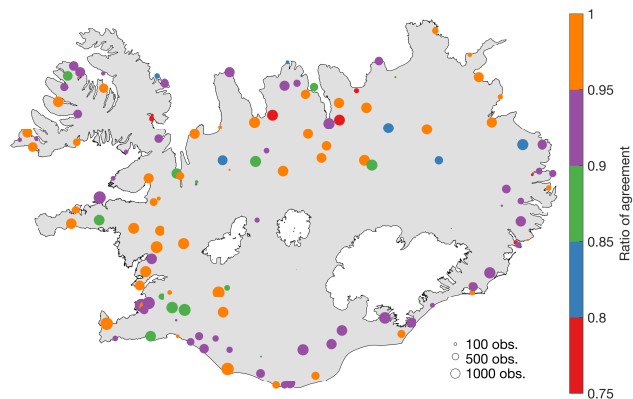

**Figure 3.** Comparison of observed snow cover and Modis daily combined snow cover. For the whole dataset the overall classification accuracy was 0.925

of snow covered pixels were correctly classified in MCDAT product while 92.7 % of snow free pixels were correctly classified. Finally for Sentinel 2 85.6 % of snow covered pixels were correctly classified in MCDAT product while 91.8 % of snow free pixels were correctly classified. Validation data from all satellites provide data over all Iceland for multiple times. Pixel density range from 110, 30 and 90 for Landsat 7, Landsat 8 and Sentinel 2, respectively. Figure 4 shows the average agreement for

5 snow covered pixels for Landsat 7, 8 and Sentinel 2 compared to the MCDAT product. Visually the agreement is good in all cases with $R^2$ values 96 %, 92 % and 95 % for Landsat 7, 8 and Sentinel 2 respectively. No clear trends or correlation can be seen between months within the year and classification accuracy. These results are in agreement with similar studies where a pixel based comparison was conducted (Huang et al., 2011; Gascoin et al., 2015).

For each Landsat 7/8 and Sentinel 2 tile a classification map was constructed. The classification maps show the agreement of

10 different satellite sources to the MCDAT product. A selected sample of the maps were manually screened to identify patterns in misclassification. The screening reveals that disagreement was mainly located at snow cover boundaries, i.e. where snow free land meets snow covered land as well as boundaries of clouds and land. Previous studies in snow covered Arctic and alpine areas have revealed a similar effect when comparing Modis to higher resolution data (Gascoin et al., 2015; Déry et al., 2005; Rittger et al., 2012). A source of misclassification has been related to effects of forested areas which should be limited

15 in Iceland due to few forested areas and sparse vegetation in general, especially at higher elevations. The effect of the Modis



**Table 2.** Confusion matrix for snow cover derived from Landsat 7, Landsat 8 and Sentinel 2 compared to Modis Aqua and Terra daily snow product combined.

|  |  | MCDAT | |
| --- | --- | --- | --- |
|  |  | No snow | Snow |
| Landsat 7 | No snow | 93.2% | 6.8% |
|  | Snow | 13.6% | 86.9% |
| Landsat 8 | No snow | 92.7% | 7.3% |
|  | Snow | 16.6% | 83.8% |
| Sentinel 2 | No snow | 91.8% | 8.2% |
|  | Snow | 14.4% | 85.6% |

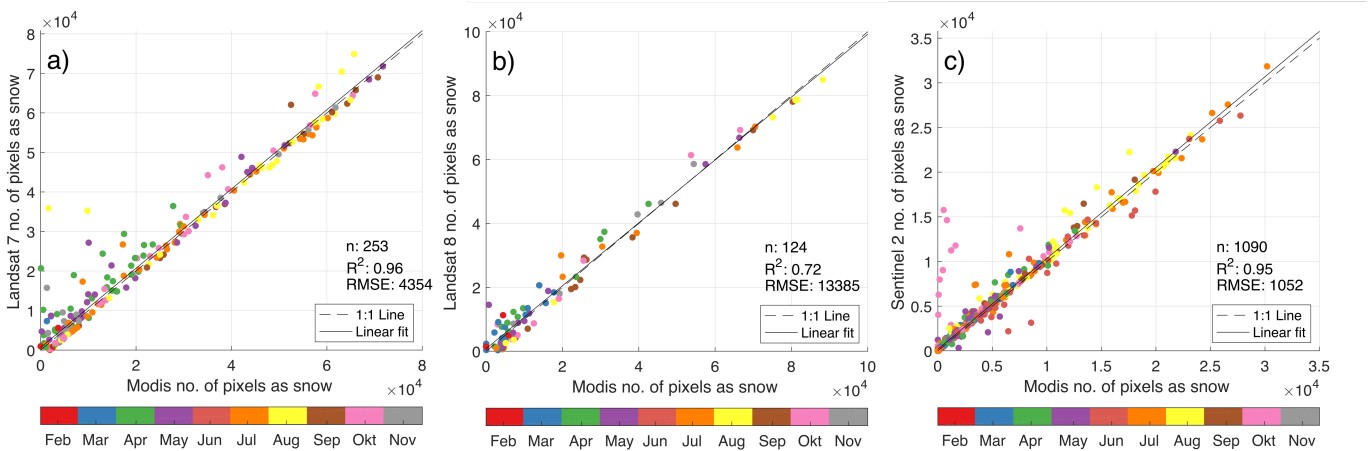

**Figure 4.** Relationship of classified snow pixels for Landsat 7, 8 and Sentinel 2 to the MCDAT product.

sensor view angle has also been identified as a source of errors where the M∗D10_L2 swath granule (source data for Modis snow products) has different boundaries producing a "bow-tie" effect which can increase misclassification (Gómez-Landesa et al., 2004).

## 4.2 Gap filling with merging and temporal filtering

5 Figures 6 and 5 show the average cloud cover frequency in Iceland based on 18 years of Modis data from 2000 – 2018 (MC-DAT). Average cloud cover for Iceland was 79 % while certain patterns are observed in the central highlands, over glaciers and in mountainous areas near the coast. In general cloud cover was less in the highlands but highest near the coast and in mountainous areas and fjords, such as Tröllaskagi, Austfirðir and Strandir. This illustrates the cloud obscurity problem for





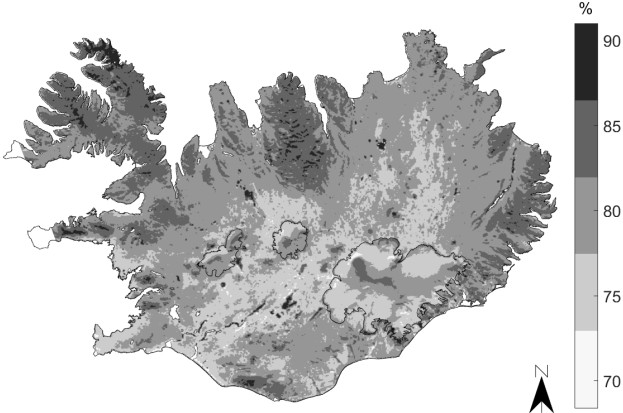

**Figure 5.** Average cloud cover over Iceland based on the MCDAT product for February to November each year from 2000 to 2018.

optical satellite remote sensing in Iceland. Figure 7 shows the results from daily merging and temporal aggregation of snow cover data. The two daily snow cover tiles from MOD10A1 and MYD10A1 had similar average cloud cover (76-78 %). Data from Aqua (afternoon overpass) showed 1.5 % average more cloud covered pixels than Terra. Merging of the Aqua and Terra daily datasets provided on average a 7 % reduction in cloud obscured pixels which was mostly related to moving cloud patterns

5  within the day. Temporal aggregation of daily merged tiles had an exponential decaying shape of unclassified pixel reduction with the highest benefit for aggregating one day. The disadvantage of aggregating more days to a center date to further reduce unclassified pixel is temporal dampening of events and rapid changes in the snow cover. For our study 3 days were aggregated to the center date, both forward and backward, meaning that for each date of aggregated data in total 7 days contributed data with priority to the most recent observations. On average the unclassified pixels were reduced from 70 % to 14 %.

In general the advantage of temporal aggregation of data is reduced cloud obscured pixels which provides a spatiotemporal continuous product. The trade-off of temporal aggregation contrasts with the dampening of the response of the snow cover to rapid melt or snowfall events. This poses a limitation on the use of the data in real time applications such as short term flow forecasting for water resources.

15  **4.3  Gap Filling with classification learners**

After applying a temporal aggregation to the data unclassified pixels still remained in the dataset. To classify the remaining pixels various classifiers were tested to assess their classification accuracy. Various configurations of classification trees, k Nearest Neighbour algorithms (fine, coarse, cubic, weighted, boosted), supportive vector machines (SVM), linear and quadratic discriminant classification learners were tested in various configurations. Overall no one method and configuration provided a

20  significant classification accuracy improvement. Average classification accuracy ranged over 90 % for all methods tested and in general had lower classification accuracy during melt season (April, May, June). The marginal best classification performance





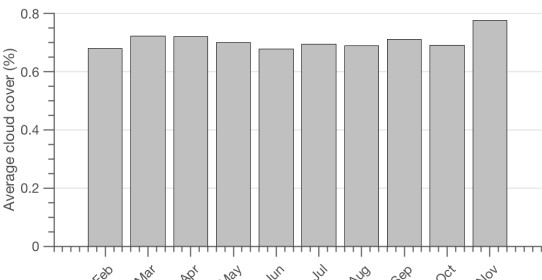

**Figure 6.** Average cloud cover distribution between months.

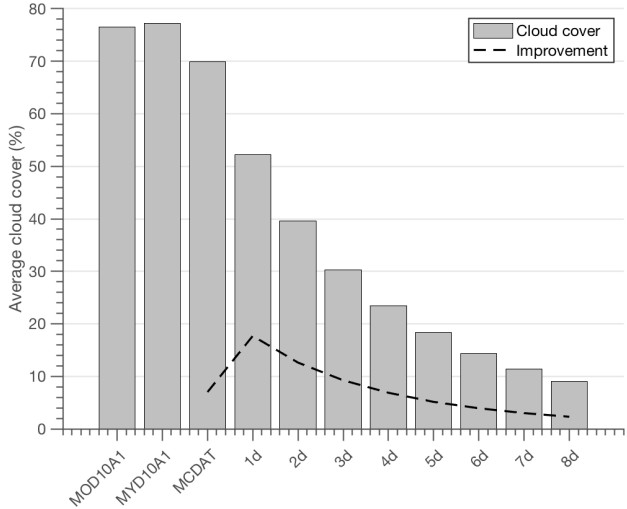

**Figure 7.** Average gap filling improvement with merging of daily data and temporal aggregation.

was by a weighted k Nearest Neighbor (wkNN) classifier which had 100 number of neighbors. The average classification accuracy for the whole dataset was 96.4 % with a standard deviation of 2.7 % and a minimum classification accuracy of 83.4 %. This was based on a 25 % withheld classified population for every date classified in every method tested. The weighted k Nearest Neighbor was selected for further use.

## 4.4 Snow cover spatial and temporal characteristics in Iceland

A daily gap-filled snow cover product was derived for Iceland based on Modis sensor bi-daily overpasses at an temporal resolution from 1.3.2000 to 30.06.2018. From the gap-filled snow cover product all water bodies and glaciers were excluded. Based on the dataset various descriptive spatiotemporal dynamics of snow cover in Iceland can be derived. The main limitation to the dataset was polar darkness during December and January that limits the continuous temporal structure of the dataset. Snow cover duration within a season is a parameter that is often used to describe characteristics of snow cover. The duration of



snow cover is a property that can be linked to many applications such as seasonal snow melt magnitude for operational water resources and length of vegetation season.

Figure 8 shows the distribution of snow cover over the whole country of Iceland from February 2000 to December 2017 as a percentage of land area in Iceland. A value of 100 % indicates that all non-glaciated land is fully covered with snow while 0 % indicates a snow free area. During late winter (February, March) most land area (>90 %) was snow covered, while during April, May and June seasonal snow cover recedes rapidly due to longer daylight hours and increasing average temperatures. Extended snow cover duration was seen in 2013, 2014, 2015 and 2016 with more than 50 % of Iceland snow covered until the end of May. Specific snow fall events can be seen in May increasing snow cover extent, but generally these events had a short impact. In the fall many events can be observed where snow cover increases in a snowfall event and then melts few days/weeks later. This shows quite well the temporal structure of Icelandic snow cover where large areas covered with snow can melt out quickly during fall and winter due to storm tracks bringing warm air masses that can both precipitate as liquid or solid precipitation. For hydropower operations in Iceland snow fall in the fall (late August, beginning September) can be a critical point in time as it can indicate lowering of inflows to reservoirs and diversions and the start of reservoir regulation season, i.e. more water is flowing out of storage than in. This is related to the influence fresh snow cover with high albedo has on the dark glacier ice in the ablation zone, reducing severely the available energy for melt.

Figure 9 shows descriptive fits for number of snow free dates (SFD), First Snow Free Date (FSFD) and Last Snow Free Date (LSFD) for the gap-filled dataset. The criteria were that the representative area had 10 % or less of the area snow covered for more than 5 consecutive days and in the case of the last snow free date the area needed to have 10 % or higher snow cover for 5 consecutive days. The number of snow free dates is the number of days between these valuse (FSFD and LSFD) annually. A commonly used valuable snow cover metric is length of snow season, i.e. the number of days where snow covers the ground. Due to polar darkness this limits the temporal continuity of the dataset during winter so length of snow season can not be described fully here.

Various studies of snow cover where polar darkness applies, a filter assuming that if a pixel has snow at the beginning of polar darkness (late November in Iceland) and the same pixel still has snow when polar darkness recedes (mid January in Iceland) it can be assumed that the snow cover is continuous for that time period (Lindsay et al., 2015; Dietz et al., 2012). In Iceland the assumption of a continuous snow season during polar darkness is feeble as winter floods can influence large areas in a single depression low event (Kundzewicz, 2012; Rist, 1990).

Figure 9a shows the first snow free date for each year in the dataset and can be related to the timing when no snow remains within an area. An expected behaviour is observed where lower elevation areas experience melt out earlier in the year than higher elevation areas. The average first snow free date for Iceland is 27th June each year with a standard deviation of 19.9 days (standard deviation is shown in parenthesis from now on). The elevation band from 0-200 m a.s.l. (25 % of Iceland) has an earlier first snow free date (7th May (+/- 13 days)) while as with higher elevations the snow cover extent is prolonged into





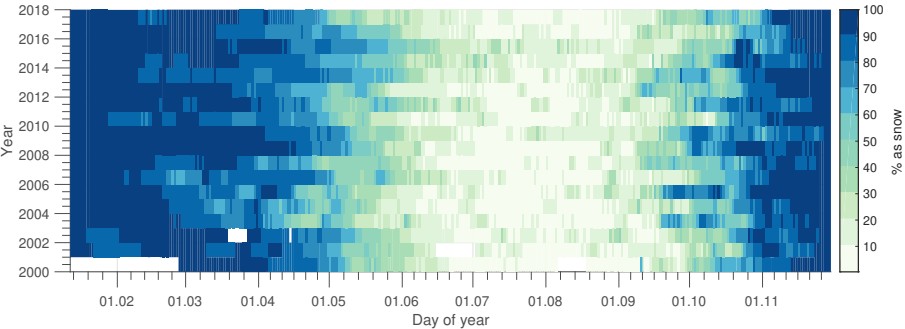

**Figure 8.** Snow cover duration (SCD) in Iceland from 1.3.2000 to 31.12.2017. 100 % indicates full snow cover while 0 % represents a snow free area. Glaciers and water bodies are not included.

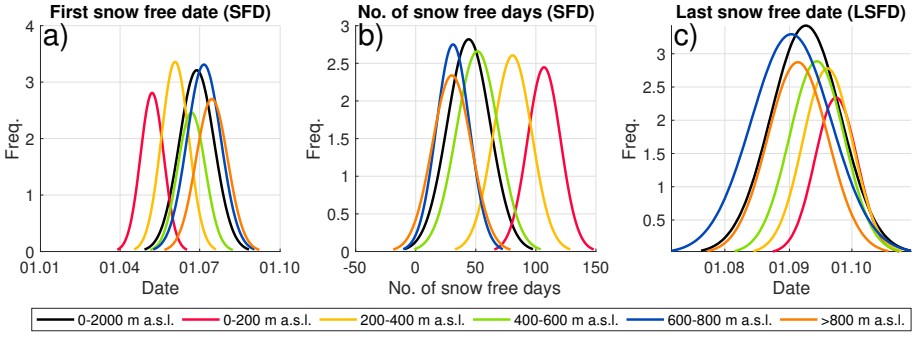

**Figure 9.** Normal distributions for extracted first snow free date, number of snow free days and last snow free date.

the summer months. Figure 9b shows the number of snow free days (SFD). For all of Iceland the number of snow free days is 44.1 days with a standard deviation of 17.8 days. For elevations from 0-200 m a.s.l. 106.8 days are snow free with a deviation of 13.7 days. For higher elevations the snow cover is more persistent and snow cover days total 80.6 days (+/- 15.9 days), 51.4 days (+/- 17.5 days) and 31.0 days (+/- 13.8 days) for 200-400 m a.s.l., 400-600 m a.s.l. and 600-800 m a.s.l., respectively.

5    For the highest elevation bands (> 800 m a.s.l.) fewer pixels are non-glaciated (see Fig. 1) with 30 (+/- 16.1) snow free days. Figure 9c shows the last snow free date (LSFD) which is on average for Iceland 8th September (+/- 16 days). This is highly influenced by snow fall events in the late fall in the highlands where snow fall events are frequently observed in late August or early September. These events will frequently melt out again which can be seen in the variable snow cover duration in Figure 8. In general higher elevations have snow fall events earlier in the fall which coincides with a later snow free date annually and

10   a fewer snow free dates, as expected.

Figure 10 (first column) shows the mean snow cover duration for pairs of months as well as for the whole period the dataset covers from February to November. Monthly averages are combined for two months, February and March, April and May, June and July, August and September, October and November. These two-month period pairs can be related to seasonality of




the snow cover where February and March represent late winter where rain on snow events or warm storms are dominating in reducing the snow cover extent. April and May represent the conventional snow melt season with snow melt commencing earlier at lower elevations and is mostly driven by gradually warming temperatures, and June and July represent the summer season where most areas are snow free except at higher elevations and glaciers. In general this also is the period when glacier

melt becomes the dominating water source in glacier fed rivers and succeeds seasonal snow melt. August and September represent late summer and early fall where highlands start to have lower temperatures, freezing during the night can be common and snow fall events are observed. October and November then represent the early winter period. In February and March land above 200 m a.s.l. is on average 100 % covered with snow with a more varying snow cover extent at lower elevations, especially near the coastline and in the southeast and southwest part. In April and May larger areas are snow free from 0 - 400 m a.s.l.

while snow cover is persistent in the highlands on higher mountains and a glacier boundaries. Snow cover has more extent in the northern part of Iceland as well in the west and east fjords. In June and July snow cover is generally at its minimum with patchy snow cover in the center highlands and on higher mountain tops and ridges, especially in the northern Westfjords, on Tröllaskagi in the north. Generally first snow is observed in August and September on glaciers and highlands with less frequent events in lower elevations in August, though often in September. On average the highlands are fully snow covered in October

and November.

Figure 10 (second column) shows changes in snow cover within each period, presented with the standard deviations of data. The data are identically to snow cover extent characteristics where there is more deviation is in the lower elevations in general during winter and moves higher in elevation during the melt season. In February and March deviation is highest

below 200 m a.s.l. and near the coast. The highest deviation is observed on the south central lowlands and along the southeast coast which relates to the governing storm track alignment during winter (Einarsson, 1984). In April and May the variability moves higher in elevation associated with seasonal snow melt and still the highest in areas in north, while east and west have less variability. In June and July the highest areas in north, east and west are melting, extending into August and September. In early winter, October and November, the snow cover in the highlands has stabilized with more variability at lower elevations.


Figure 10 (third column) shows trends in snow cover within each period. In February and March the average trend is close to zero with some areas (red) where snow cover extent recedes over the period. For April/May, June/July and August/September the average trend for each period is positive, indicating that the snow cover extent was spanning a longer time, i.e. snow cover was extending further into the spring and summer months. For early winter, October and November, the average trend was

negative meaning that snow cover was on average less, especially in the east and north. Further details of this are shown in Figure 11 where monthly mean snow cover extent was calculated for all years and the data were fitted linearly.

As previously mentioned the dataset only spans 18 years so statistical interpretation, such as trends should be treated with care. To evaluate if these trends are significant a linear trend test and a Mann Kendall test was performed on monthly mean snow cover extents for $\alpha$ equal to 0.05. The Mann-Kendall test was a non-parametric test to identify trends in data over time

where no assumption of normality was required (Mann, 1945; Helsel and Hirsch, 2002). Results indicate that the observed



trends in the data are insignificant for all months except June, tested both with the Mann-Kendall test as well as the linear trend test. As identified visually, of the data in Figure 11 for May, June and July the steep trend was governed by snow cover extent in 2013, 2014 and 2015 which were abnormal years compared to previous years with below normal spring and summer temperatures which resulted in an extended length of the seasonal snow cover season which also reflected in positive mass

balance of all Icelandic glaciers for the first time in over 20 years (Pálsson and Gunnarsson, 2016b, a; Þorsteinsson et al., 2017). Similarly, a slightly negative trend for October and November was calculated from a linear fit and was also governed, less though, by extended liquid precipitation events in these months in 2014, 2015 and 2016. A non-statistical parameter, $\Delta y$, was calculated to represent the average change over the period. This is merely the average slope of the linear fit but provides insight into the average characteristic of snow cover trend.

Figure 12 shows average snow cover extent for different elevation bands for Iceland. The influence of elevation on the average snow cover extent is a strong controlling factor where large areas over 800 m a.s.l. retain the snow cover throughout the summer. During spring (April/May) a strong increase in snow cover extent was observed between 0-200 m a.s.l. and for the evaluation bands above 200 m a.s.l. This is consistent with results from Björnsson et al. (2018) where the annual average

0°C isotherm is defined ranging from 200-300 m a.s.l. During winter, elevation over 600 m a.s.l. are mostly fully covered with snow. The snow melt season occurs in April to July depending on elevation. In the fall a strong increase was also observed between September and October. Figure 13 shows the distribution of snow cover within the four main aspect classes (N, E, W, S). During February, March, April, October and November it shows that the snow cover tends to persist longer on the north-, west-, and east-facing slopes. During summer (June, July, August) this effect is less dominant. This is consistent with expected

snow pack energy balance where in general north-facing slopes receive less solar radiation for melt while east-, and west-facing slopes are exposed to a similar amount of solar radiation at different times of the day (west facing in the afternoon and east facing in the morning).

## 5 Conclusions

In this study, a gap-filled satellite observed snow cover was produced from daily Modis Aqua/Terra observations with duration

from early 2000 until 2018 at a 500 m spatial resolution. Overall a good agreement was found between the daily combined Modis Terra/Aqua dataset and the validation datasets from Landsat 7/8, Sentinel 2 and in-situ observations in Iceland. The Landsat and Sentinel data showed that boundary artefacts were present in the Modis product at cloud/land boundaries while no seasonal patterns of agreement were found when validating alternative remotely sensed products.

Average cloud cover in Iceland is high (75 % average) providing a significant limitation to the application of Modis data and all optical remote sensing instruments. No significant temporal patterns were found in cloud cover while the central highland in general has lower average cloud cover. This was addressed with temporal aggregation of data where the tradeoff from temporal aggregation (7 days) could have implications for hydrological applications of the dataset where onset of melt and melt events





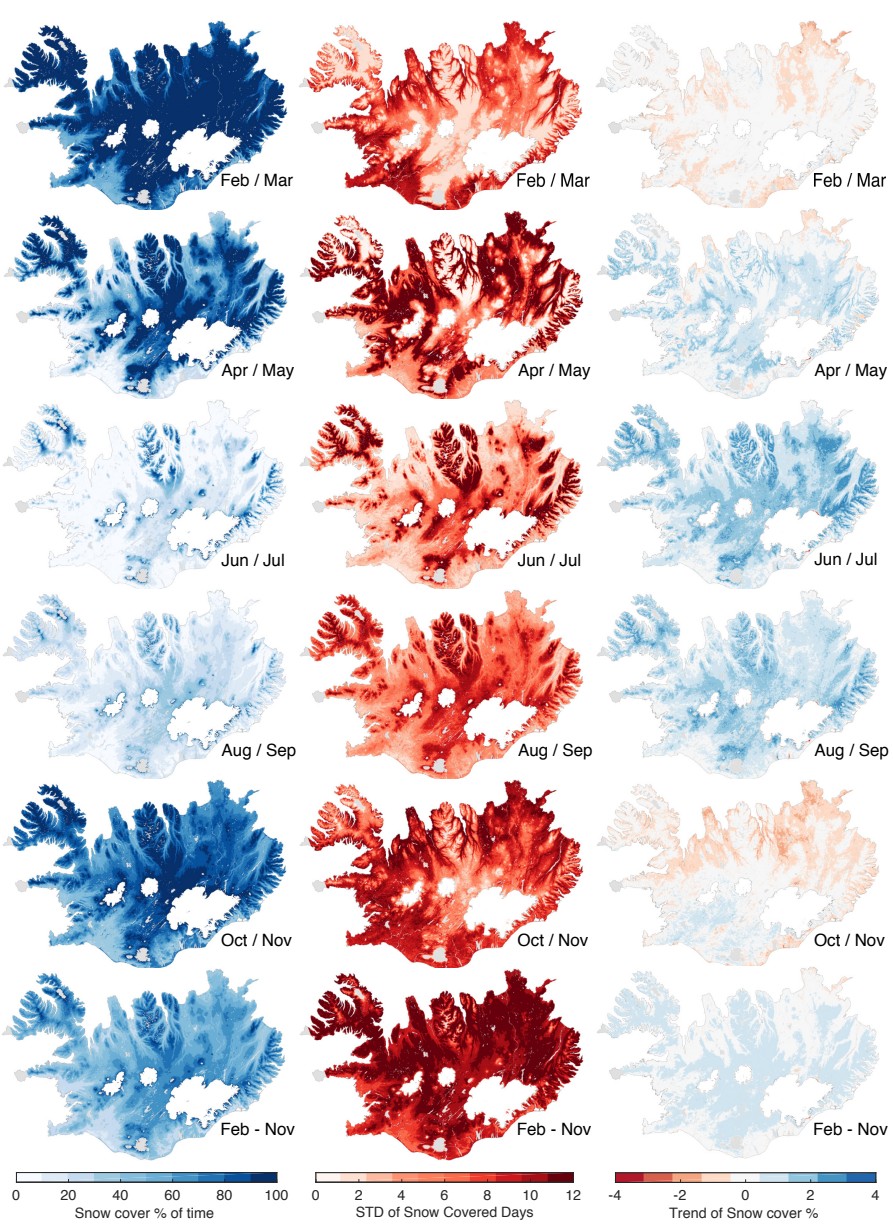

**Figure 10.** First column: mean snow cover duration as percentage of time for each period. Second column: Standard deviation of days for each period. Third column: Mean trend in snow cover duration as percentage of time for each period. Rows represent different combinations of monthly values and the bottom row is for the whole period from February to November.



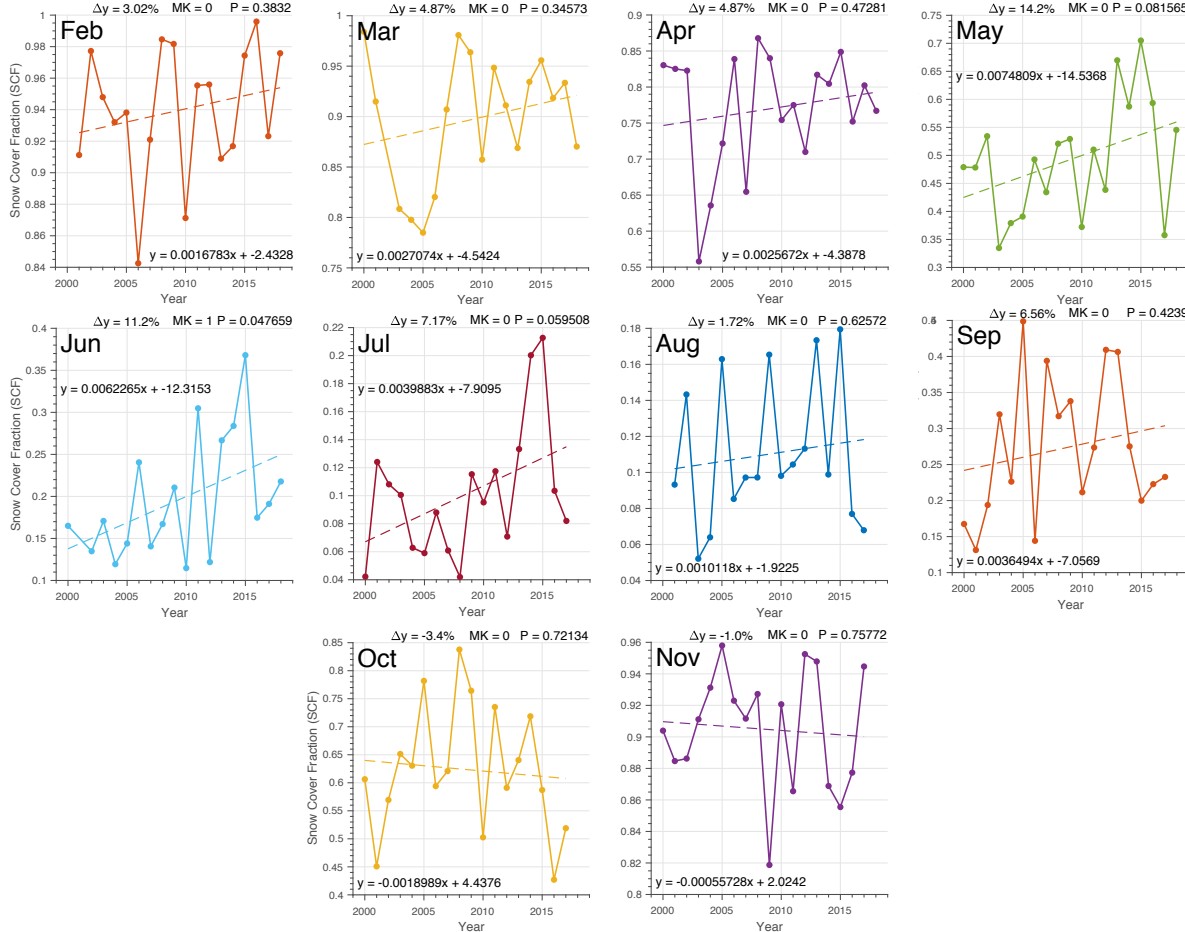

**Figure 11.** Monthly mean results for snow cover. For each month mean snow cover extent is calculated and a linear fit applied. The results from the Mann-Kendall test are shown as 1 or 0 where 1 indicates a significant change in trend. A linear equation shows the results for a fitted linear model to the dataset. $\Delta y$ is a non-statistical parameter and shows the average linear slope of the trend.

could be retained or smoothed out of the product. This was also a limitation for identifying rain on snow events during winter.

Availability of Modis data during Polar darkness was also a temporal limitation for the dataset. From late November to mid January no data were available which limits the application of the dataset to identify rain on snow events that can cause
5  flooding and deplete areas of snow pack. Due to the dynamics of Icelandic snow during winter, especially at lower elevations, this is challenging to solve without combining other data sources such as snow models or other sources of remote sensing, for example synthetic aperture radar such as ESAs Sentinel 1 which has a frequent overpass over Iceland.

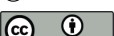



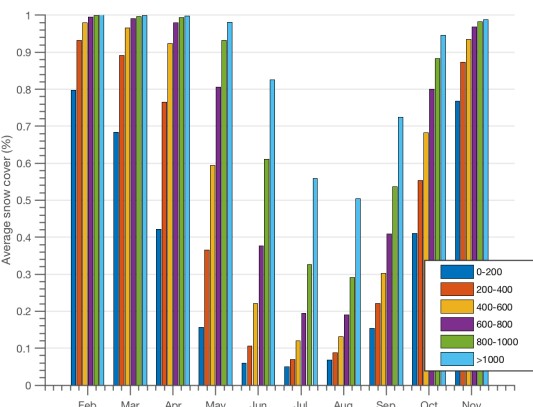

**Figure 12.** Average elevation distribution of snow cover for 2000 - 2018. Fractions of area for each elevation bands are 23.9 % for 0-200 m a.s.l., 17.5 % for 200-400 m a.s.l., 21.7 % for 400-600 m a.s.l., 18.4 % for 600-800 m a.s.l., 8.2 % for 800-1000 m a.s.l. and 9.9 % for elevations over 1000 m a.s.l.

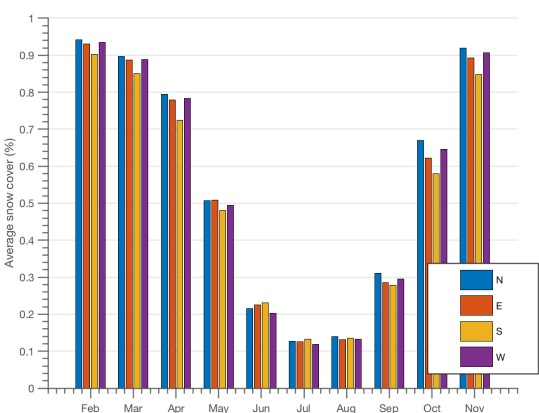

**Figure 13.** Average aspect distribution of snow cover for 2000 - 2018. Fractions of area for each aspect are 22.8 % for 0 - 90°, 23.8 % for 90-180°, 23.4 %for 180-270 °and for 270-360° 26.6 %

The changes over time (trend) analyzed for the 18 years showed a slight increase in average snow cover. This aligns well with observations of winter mass balance of Icelandic glaciers with a slight significant positive trend for the past 20 years (Pálsson and Gunnarsson, 2016c) as well as an observed precipitation increase (Björnsson et al., 2018).

The gap-filled snow cover product provides a useful tool to monitor and analyze inter-annual variability and long term
5   trends in snow cover in Iceland. The methodology applied here can be applied to other satellite sensors such as Sentinel 3 or the Visible Infrared Imaging Radiometer Suite (VIIRS) to extend the temporal range of data beyond the Modis mission.





*Code and data availability.* Code used in the project to process data is available at: https://github.com/andrigunn/isca. Modis data are available from https://nsidc.org/data/, Sentinel 2 data are available at https://scihub.copernicus.eu/dhus/ and Landsat 7 and 8 data at https://earthexplorer.usgs.gov. Data set tiles, paths and version numbers are defined in Section 2. Geospatial data for Iceland are available from the National Land Survey of Iceland at https://atlas.lmi.is/LmiData/index.php. Observations of snow cover from manned IMO sites are available upon request to fyrir-spurnir@vedur.is.

*Author contributions.* AG conceived and designed the study, performed the analyses, and prepared the manuscript. SMG and ÓGBS contributed to the study design, interpretation of the results, and writing of the manuscript.

*Competing interests.* The authors declare that they have no conflict of interest.

*Acknowledgements.* We would like to thank Professor Jessica D. Lundquist, University of Washington, for discussion and valuable feedback
10 during the design of the study. Special thanks to Helgi Jóhannesson, Project Manager at Landsvirkjun, for providing constructive feedback during review of the manuscript. The Valle Scholarship and Scandinavian Exchange Program at University of Washington is also thanked for financial support during the academic year 2017-2018 at the University of Washington.





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
