# Peer review of "Icelandic Snow Cover Characteristics derived from a gap-filled MODIS Daily Snow Cover Product"

_Hydrology and Earth System Sciences, 2019_

## Referee Comment (RC1) · Anonymous Referee #1 · 28 Mar 2019

General comments:

This study presents a complete characterization of snow cover characteristics in Iceland based on a remote sensing product. The authors present the methods used to obtain a gap-filled dataset of snow cover based on a Moderate resolution Imaging Spectroradiometer (Modis), as well as the validation of this product with other satellite data and in-situ observations. Although the context of the novel methods and satellite products is explained well, the study could be better placed in the context of the importance of snow cover studies under climate change. After a successful validation the dataset, the authors analyse the characteristics of snow cover extent and duration over

the whole of Iceland. Despite the limitations of satellite products in polar latitudes due to polar darkness and clouds, a thorough monthly analysis of snow cover from year 2000 to 2018 is presented. A trend analysis for such a short period is also done. I do not see a major issue on that, since the aim of the paper is not to present a trend analysis and the conclusions are not weakened, but some parts of it require a clarification or a rephrasing of the text.

A data product of this spatio-temporal characteristics over a highly snow dependent region such as Iceland did not exist before and is therefore an advance for snow and hydrological studies. The methods could be further used in other snow dependent regions where the satellite products have coverage, and therefore publication of this article would promote scientific progress. In addition, the article falls well in the scope of the journal since such a complete snow cover product might be of interest for catchment and water cycle studies over Iceland and for operational use as e.g. in the prediction of hydropower generation based on snowmelt. The text follows a logical story and in general is well written, with all sections explained thoroughly in detail.

I suggest accepting the paper for publication after improving some minor issues of the trend analysis, as well as the presentation of some results and figures and the writing of some parts of the text, as I detail below:

Specific comments (minor issues):

Introduction:

- In the first paragraph of the introduction, more references could be used to place the reader in the context of snow cover studies based on satellite products ( https://doi.org/10.5194/tc-5-219-2011, https://doi.org/10.1029/2012GL053387 , https://doi.org/10.1175/2010JCLI3644.1 ) and the importance that these have because continental perspectives on snow cover changes due to climate change, based on in-situ observations, are only starting to become available and data is scarce (especially over Iceland) https://doi.org/10.1029/2018GL079799

[Figure]

Data and Methods:

- In both sections the order of "in-situ data" "Modis data" and "Landsat-Sentinel data" should be the same, it is now different and confuses the reader. I suggest 3.1 to be "in-situ" data, 3.2 to be "Modis" and 3.3 "Sentinel".

- The methods section is quite technical and therefore could highly benefit from a schematic Figure representing the types of data and the types of processing of the data, in the form of a "flowchart". This would help the reader follow better the whole methods process.

Results:

- Table 1: Can you please provide an explanation or at least hypothesis on why agreement is lower when observations show snow than when they show no snow? For instance, if satellite products were to confuse a cloud with snow, that would lead to a lower agreement when observations show no snow than when they show snow. The difference is big so there should be some reason as for instance that snow is not deep enough for the satellite product to be detected?

- Figure 4: The display with a different colour for every month gives no additional information since nothing can be seen from the colours (except for a few clusters). I suggest that a correlation between landsat and modis is computed for every month and then presented in a table. This would potentially identify in which months MODIS performs best or worse and would give a more complete validation.

- Some statements about Figure 10 and 11 on trend analysis should be treated with more care. Page 15, lines 25-30: "In February and March [...] some areas where snow cover extent recedes over the period"; The statement is too strong, since looking at Figure 10 right column Feb-Mar, trends over almost the whole of Iceland are <1%, which in case of being % change over two months would represent a change of 0.6 days. This rate, considering standard deviations are generally higher than 5 (as seen

in the middle column) is highly insignificant. A similar statement is written for Oct-Nov, with trends that are generally smaller than 1.5%, and when considering the whole year (Feb-Nov). I suggest decreasing the strength of the statements or showing the trends differently, for instance computing the trend divided by standard deviation (in days, not percentage). This would should on the map to what extent these changes are significant, and might support the significant increasing trend obtained for June when computing the snow covered fraction (Figure 11). Regarding these increasing trends for May, June and July, the significance for such a short period could be explained by the 3 abnormal years in 2013, 2014 and 2015. Although this is stated in the results, I suggest that this is mentioned in the conclusion. Moreover, the conclusion should indicate that an increase is only observed in June or spring, as it is well indicated in the abstract (Page 19, line 1): "The changes over time (trend) analysed for the 18 years showed a slight increase in average snow cover in spring, probably driven by 3 abnormally cold years in 2013, 2014 and 2015. This aligns . . ."

Technical corrections/clarifications:

- p.2 L-8: What is the order of the references? It is not alphabetical and not old to new.

- P.2 L-21: Remove extra brackets in (Fig 1,2)

- P.4 L-29: I suggest changing "main objective" for "aim", since after this sentence a first and second objective are presented.

- P.5 L-14: Join the two references.

- P.5 L-19: Please explain what tile h17v02 is and where it comes from

- P.7 L-4: Abbreviation MCDAT appears here for the first time but it is not explained, please provide the full name for it.

- P.7 L-7: What is the best observation of the day? Can there be two best? What happens then?

- P.8 L-20: Are the numbers correct? 213.011 matches out of 585.800 is less than 50% accuracy.

- P.8 L-25: "at the bottom"

- Figure 3: Please change the colour scale to a continuous one, otherwise it is difficult to read the map.

- I suggest merging Figures 5 and 6.

- P.12 L-9: While the text indicates that December and January are not available, Figure 8 shows 11 months of data. How is that possible?

- Figure 9: Please increase figure size if possible.

- P.15 L-29: This result contrasts with other studies showing a shortening of melt season and earlier onsets https://doi.org/10.1175/2010JCLI3644.1

―――――――――――――――――

---

## Referee Comment (RC2) · Anonymous Referee #2 · 30 Apr 2019

General Comments Cloud cover and persistence is a substantial obstacle for monitoring snow presence/absence, especially in locales with abundant clouds. The manuscript describes a methodology for a gap-filling approach to remedy the issue. The resultant product is compared against ground observations (snow absence/presence) and coincident higher-resolution imagery. The product is analyzed for landscape characteristics and temporal trends. The manuscript presents an advance in analysis of snow duration and site characteristics, especially for such a heavily cloud-dominated Iceland snow pack. Analyses such as this, performed at moderate resolution, add insight into snow variability and temporal trends that are useful and creative. As presented, the conclusions reached are carefully supported by the methods

and results, and the structure of the manuscript makes it relatively easy to follow the authors' lines of thought. The paper is referenced well overall. The manuscript is hindered most by its writing, rather than its content. Substantial effort to make the writing efficient, precise, and concise throughout would go a long way to improve its clarity and make it accessible and elegant. In some places there is too much description that can be reduced. Scientifically, the paper is solid and informative. The manuscript can be improved by addressing important work done on published snow cover trends in the Northern Hemisphere and Iceland (e.g., Dietz et al.) in the Discussion.

Specific Comments Pg. 1, line 20-21. "... and low thermal conductivity which is dominating for the growing season length of vegetation and plants (Keller et al., 2005)." How is snow's thermal conductivity related to growing season length, let alone a dominant for determining growing season length? Thermal conductivity is indeed important for flora, fauna, and soils in winter; but what ties are there between thermal conductivity of snowpacks and growing season length? In addition, why "vegetation and plants" in the sentence? Isn't vegetation comprised of plants?

The description of Icelandic land cover is especially relevant (Pg. 2, lines 12-14). The sparse and bare can be envisioned, but what is meant by "semi-natural" vegetation? Figure 2. By the time the reader arrives at this Figure, there has been no introduction of what size pixel is being referenced. Perhaps hectares or sq. km would be more useful as a y-axis variable.

Pg. 3, lines 9-11 is confusing: "A system of reservoirs and diversions store melt water during the spring freshet which generally consists of a seasonal snow melt period (April - June), a glacier melt period (June - September) and precipitation in the fall (August - October)." If you replace "spring freshet" with "year," it makes sense, but I'm not sure if this captures the intended point.

Pg. 5, lines 23-26. At times there is too much detail in the manuscript, and this is a good example of text that can be cut. "The data were downloaded from the United

States Geological Survey (USGS) (https://earthexplorer.usgs.gov/) using bulk download utilities. Landsat scenes that cover Iceland are numbered from 224-13 to 216-13, 224-14 to 215-14, 223-15 to 215-15 and 219-16 to 216-16 in the Worldwide Reference System 2 (WRS2), a total of 32 Landsat footprints (USGS, 2018)." Just referencing the source of the data and website in the previous sentences should suffice.

Organization. In Section 2 (Data), the ground observations are described first. In Section 3 (Methods), the Landsat/Sentinel data are described first. Perhaps 3.2 in-situ data processing, could be moved to 3.1 to maintain that structure? It goes back to ground observations first in Results (4.1.1)

Pg. 6, lines 22-25. This part discusses resampling for sentinel data, but there is no corresponding parallel description of this for Landsat data in the first paragraph of Section 3.1. More important, what sort of resampling was used to shift the pixels from 30 and 20 m resolution to 500 m? What impacts are expected from the scale disparity going from 30/20 to 500 m? Of the pixels resampled, how much snow-covered classifications went to snow-covered areas or vice-versa? To elucidate, does snow cover largely disappear at once, or do lingering drift areas remain? What sensitivity is in the snow-classification developed from MODIS to how much pixel area is snow covered before that threshold of snow presence/absence is crossed for Iceland? Some discussion of these scale issues and inherent differences would be appreciated. Later on in the manuscript, the line (Pg. 9, lines 11-12), "The screening reveals that disagreement was mainly located at snow cover boundaries, i.e. where snow free land meets snow covered land as well as boundaries of clouds and land," is intriguing. It seems like more should be said about these boundaries and what is and isn't captured in the approach and validation with higher-resolution data.

Pg. 8, lines 3-5. "To classify the remaining unclassified pixels information about location (Latitude, Longitude), elevation and aspect to account for earlier melting of south facing slopes are derived to apply for a gap filling algorithm." This is unclear.

[Figure]

Figure 3. The color bars for the ratio of agreement aren't intuitive. Orange and Red bracket purple, blue, and green. The error magnitude would be better understood if the color scheme made a more natural color progression from high (warm/cold) to low (cold/warm).

Pg. 9, lines 3-4. "Pixel density range from 110, 30 and 90 for Landsat 7, Landsat 8 and Sentinel 2, respectively." This sentence is unclear.

Pg. 9, lines 5-6. "Visually the agreement is good in all cases with R2 values 96 %, 92 % and 95 % for Landsat 7, 8 and Sentinel 2 respectively." This statement isn't in agreement with Figure 4B, where the R2 is listed as 0.72 and the MSE seems high.

Pg. 11, lines 16. "After applying a temporal aggregation to the data unclassified pixels still remained in the dataset." Please tell us more about that here; how many? What percentage?

Figure 8. Wait, on page 12 line 9 we learned Nov-January data were not there due to darkness, and the Figure presents 12 months in a year on the X-Axis. Day of year on the X-Axis should be in DOY, not months. On the Y-Axis, why not use one tick per year instead of 0.5 year?

It would be more useful if this work were placed in a similar context with published analyses of snow cover trends. There's no discussion on volcanic impacts on snow duration. There are no contrasts provided with other published results/trends for snow cover, even at Northern Hemisphere scales. Claiming that increased glacial mass balance in Iceland is interesting, but may not be identical to what is being observed/measured in this project.

Figure 10 is interesting. It would be helpful to add a small black line to separate the Feb-Nov full dataset analyses from the bi-monthly comparisons comprising the top.

Technical Corrections The paper could be shortened a bit with increased efficiency. "Modis" should be "MODIS" throughout.
The manuscript has a comma shortage, and there are a number of single-sentence redundancies throughout where identical words are used repeatedly in the same sentence or adjacent sentences.

Pg. 2 Line 20, "higher altitudes" could be "high-altitude"

Figure 1. The green markers are hard to see on the dark gray background.

Need a "growing" between "vegetation" and "season" (Pg. 13, line 2)

---

## Author Response (AR1)

[revised manuscript text omitted]

This study presents a complete characterization of snow cover characteristics in Iceland based on a remote sensing product. The authors present the methods used to obtain a gap-filled dataset of snow cover based on a Moderate resolution Imaging Spectroradiometer (Modis), as well as the validation of this product with other satellite data and in-situ observations. Although the context of the novel methods and satellite products is explained well, the study could be better placed in the context of the importance of snow cover studies under climate change. After a successful validation the dataset, the authors analyse the characteristics of snow cover extent and duration over the whole of Iceland. Despite the limitations of satellite products in polar latitudes due to polar darkness and clouds, a thorough monthly analysis of snow cover from year 2000 to 2018 is presented. A trend analysis for such a short period is also done. I do not see a major issue on that, since the aim of the paper is not to present a trend analysis and the conclusions are not weakened, but some parts of it require a clarification or a rephrasing of the text. A data product of this spatio-temporal characteristics over a highly snow dependent region such as Iceland did not exist before and is therefore an advance for snow and hydrological studies. The methods could be further used in other snow dependent regions where the satellite products have coverage, and therefore publication of this article would promote scientific progress. In addition, the article falls well in the scope of the journal since such a complete snow cover product might be of interest for catchment and water cycle studies over Iceland and for operational use as e.g. in the prediction of hydropower generation based on snowmelt. The text follows a logical story and in general is well written, with all sections explained thoroughly in detail. I suggest accepting the paper for publication after improving some minor issues of the trend analysis, as well as the presentation of some results and figures and the writing of some parts of the text, as I detail below.

**Author response #1:**

First, we would like to thank Anonymous Reviewer #1 for very useful comments and a general positive feedback about our submitted manuscript.

**Reviewer #1 Comment #1**

**Introduction:**

In the first paragraph of the introduction, more references could be used to place the reader in the context of snow cover studies based on satellite products ( https://doi.org/10.5194/tc-5-219-2011, https://doi.org/10.1029/2012GL053387 , https://doi.org/10.1175/2010JCLI3644.1 ) and the importance that these have because continental perspectives on snow cover changes due to climate change, based on insitu observations, are only starting to become available and data is scarce (especially over Iceland) https://doi.org/10.1029/2018GL079799

**Author response #1:**

More discussion and details will be added to a modified manuscript in the context of snow cover studies based on satellite products as suggested above. This has also been pointed out by reviewer #2.

**Author final response #1:**

The following text has been added:

In the Introduction section

In the Northern Hemisphere the spring snow cover extent has decreased significantly, influencing the dynamics of spring melt intensity and timing in recent years (Adam et al., 2008; Barnett et al., 2005; Choi et al., 2010; Hori et al., 2017). Various studies using remotely sensed data, observations and climate models unanimously agree that on the Northern Hemisphere scale snow cover extent and duration is receding by 2.5 to 10 days/decade depending on the study period (Eythorsson et al., 2019; Fontrodona Bach et al., 2018; Choi et al., 2010; Hori et al., 2017; Wang et al., 2018; Liston and Hiemstra, 2011). On regional scales snow cover changes can vary depending on local climatology and its variability. Future projections with warming trends predict less precipitation to fall as snow and snow melt to occur earlier in spring, affecting runoff and water resources downstream (Vaughan et al., 2013; IPCC, 2013).

In the Conclusion section:

These results are consistent with previous findings that suggest that an slight increase in snow cover extent/area is observed in Iceland (Eythorsson et al., 2019;Wang et al., 2018) even though a general decreasing snow cover extent/area and shortening of the melt season in the Northern Hemisphere is reported in many other studies (Choi et al., 2010; Hori et al., 2017).

Another influencing factor for onset of melt and melt enhancement is radiative forcing by light-absorbing particles (Painter et al., 2018; Skiles et al., 2018). Due to frequent volcanic eruptions in Iceland volcanic ash and tephra can be transported great distances (Gudmundsson et al., 2012; Júlíus Sólnes et al., 2013). The volcanic eruptions in Eyjafjallajökull in 2010 and Grímsvötn in 2011 took place in April and May, respectively, and are within the MODIS data period (2000 - 2018). Figure 9 and 12 show no clear sign of melt enhancement for spring 2010 and 2011 although albedo and summer melt of Icelandic

**Reviewer #1 Comment #2**

In both sections the order of "in-situ data" "Modis data" and "Landsat-Sentinel data" should be the same, it is now different and confuses the reader. I suggest 3.1 to be "in-situ" data, 3.2 to be "Modis" and 3.3 "Sentinel".

**Author response #2:**

We agree and will revise the text accordingly. This has also been pointed out by reviewer #2.

**Author final response #2:**

Order has been updated in Data and Methods. In situe, Modis and Landsat/Sentinel

**Reviewer #1 Comment #3**

The methods section is quite technical and therefore could highly benefit from a schematic Figure representing the types of data and the types of processing of the data, in the form of a "flowchart". This would help the reader follow better the whole methods process.

**Author response #3:**

We have a draft of a figure showing in detail how the classification process is undertaken and another one where the MODIS data flow is shown. Based on those we will add a figure that shows the main processing steps of the data structure including temporal merging, daily aggregation and cap filling procedures.

**Author final response #3:**

A simple schematic figure has been added that shows the main processing steps

**Reviewer #1 Comment #4**

**Results:**

Table 1:

Can you please provide an explanation or at least hypothesis on why agreement is lower when observations show snow than when they show no snow? For instance, if satellite products were to confuse a cloud with snow, that would lead to a lower agreement when observations show no snow than when they show snow. The difference is big so there should be some reason as for instance that snow is not deep enough for the satellite product to be detected?

**Author response #4:**

Possible explanation for a lower agreement could be the mismatch of pixel boundary extent compared to the manned observation "extent". This relates to many of the in situ snow cover sites are located within or close to cities and small municipals where buildings, roads and other civil structures could influence the NDSI value from MODIS towards classifying the pixel not snow covered while the manned observation would classify the site as snow covered. The lower agreement could also be attributed towards the different observation methods as the manually observed snow cover is based on a manned observation from the ground while MODIS provides an areal average of the NDSI value within each pixel.

Possible explanations for a higher agreement for no snow classification over snow classification (Table 1) could be related to that many of the in situ snow cover sites are located within or close to cities and small municipals where buildings, roads and other civil structures could influence the NDSI value from MODIS towards classifying the pixel not snow covered while the manned observation would classify the site as snow covered.

**Reviewer #1 Comment #5**

Figure 4:

The display with a different colour for every month gives no additional information since nothing can be seen from the colours (except for a few clusters). I suggest that a correlation between landsat and modis is computed for every month and then presented in a table. This would potentially identify in which months MODIS performs best or worse and would give a more complete validation.

**Author response #5:**

The authors feel that the figure shows the correlation nicely but agree that the coloring of month do not add any information. Colors will be removed.

**Author final response #5:**

Figure has been updated with no colors and monthly classifications

**Reviewer #1 Comment #6**

Some statements about Figure 10 and 11 on trend analysis should be treated with more care. Page 15, lines 25-30: "In February and March [. . .] some areas where snow cover extent recedes over the period"; The statement is too strong, since looking at Figure 10 right column Feb-Mar, trends over almost the whole of Iceland are in the middle column) is highly insignificant.

A similar statement is written for Oct-Nov, with trends that are generally smaller than 1.5%, and when considering the whole year (Feb-Nov). I suggest decreasing the strength of the statements or showing the trends differently, for instance computing the trend divided by standard deviation (in days, not percentage). This would should on the map to what extent these changes are significant and might support the significant increasing trend obtained for June when computing the snow covered fraction (Figure 11).

**Authors response #6:**

The authors agree that this needs to be clarified. Suggested change is as follows:

Sentence is:

average trend is close to zero with some areas (red) where snow cover extent recedes over the period. For April/May, June/July and August/September the average trend for each period is positive, indicating that the snow cover extent was spanning a longer time, i.e. snow cover was extending further into the spring and summer months. For early winter, October and November, the average trend was negative meaning that snow cover was on average less, especially in the east and north. Further details of this are shown in Figure 11 where monthly mean snow cover extent was calculated for all years and the data were fitted linearly."

**Authors modification #6:**

"Figure 10 (third column) shows trends in snow cover within each period. In February and March the average trend is close to zero **(insignificant for all areas).** For April/May, June/July and August/September the average trend for each period is positive, indicating that the snow cover extent was spanning a longer time, i.e. snow cover was extending further into the spring and summer months. For early winter, October and November, the average trend was **slightly** negative meaning that snow cover was on average less, especially in the east and **north in the order of 0.4 to 3 days**. Further details of this are shown in Figure 11 where monthly mean snow cover extent was calculated for all years and the data were fitted linearly."

**Authors final modification #6:**

"Figure 10 (third column) shows trends in snow cover within each period. In February and March the average trend is close to zero **(insignificant for all areas).** For April/May, June/July and August/September the average trend for each period is positive, indicating that the snow cover extent was spanning a longer time, i.e. snow cover was extending further into the spring and summer months. For early winter, October and November, the average trend was **slightly** negative meaning that snow cover was on average less, especially in the east and **north in the order of 0.4 to 3 days**. Further details of this are shown in Figure 11 where monthly mean snow cover extent was calculated for all years and the data were fitted linearly."

**Reviewer #1 Comment #7**

Regarding these increasing trends for May, June and July, the significance for such a short period could be explained by the 3 abnormal years in 2013, 2014 and 2015. Although this is stated in the results, I suggest that this is mentioned in the conclusion. Moreover, the conclusion should indicate that an increase is only observed in June or spring, as it is well indicated in the abstract (Page 19, line 1): "The changes over time (trend) analysed for the 18 years showed a slight increase in average snow cover in spring, probably driven by 3 abnormally cold years in 2013, 2014 and 2015. This aligns . . ."

**Authors response #7:**

The authors agree that this needs to be clarified. We suggest modifying the sentence (P19, Line 1) in the Conclusion to:

**Authors modification #7:**

in spring, likely driven by the three cold Springs in 2013, 2014 and 2015 and extended liquid phase precipitation in the fall for the same years.

**Authors final modification #7:**

"The changes over time (trend) analysed for the 18 years showed a slight increase in average snow cover in spring, likely driven by the three cold Springs in 2013, 2014 and 2015 and extended liquid phase precipitation in the fall for the same years.

**Reviewer #1 Comment #8**

- P.5 L-19: Please explain what tile h17v02 is and where it comes from

**Authors response #8:**

We suggest expanding the last sentence in 2.2 Modis Snow Cover data to:

"Data from NSIDC are gridded using in the MODIS Sinusoidal Tile Grid system which covers approximately an area of 1200 km by 1200 km with a nominal 500 m spatial resolution. Tile h17v02 was used in this project as it covers all the central highlands in Iceland and leaves out only a small portion of the west Snæfellsnes Peninsula and the Westfjords (dataset citation)."

**Authors final response #8:**

We suggest no addition to the text as the reviewers have commented that at times there is too much technical detail in the manuscript, and this is a good example of text that can be cut.

**Reviewer #1 Comment #9**

- P.7 L-4: Abbreviation MCDAT appears here for the first time but it is not explained, please provide the full name for it.

**Authors response #9:**

We will add the following:

MCDAT (MODIS Combined Data for Aqua and Terra)

**Authors final response #9:**

The following has been added:

… used for further processing is named MCDAT (MODIS Combined Data for Aqua and Terra).

**Reviewer #1 Comment #10**

- P.7 L-7: What is the best observation of the day? Can there be two best? What happens then?

This is an internal processing step at NSIDC so processing details are sparse but the main criteria is based on that each observation represents the best sensor view of surface in the cell based on solar elevation, distance from nadir, and cell coverage. Iceland having a high latitude has multiple daily overpasses by both satellites that are merged. We find it very unlikely that two observations can have the same quality (be best) as solar zentih angle will always be different in one data set even if the cloud cover (cell coverage) and distance from nadir would be the same. No change to the text in the manuscript is needed.

**Authors final response #10:**

No changes are made to the manuscript

**Reviewer #1 Comment #11**

- P.8 L-20: Are the numbers correct? 213.011 matches out of 585.800 is less than 50% accuracy.

**Authors response #11:**

This is correct. The context is that out of 585.800 available manned in situ observations there are 213.011 instances where a daily match can be found from MODIS, that is this relates to cloud cover over the in situ observation site. This aligns well with the high average cloud cover over Iceland, where only 30-40% of the time a Modis observations is available for a manned observation.

**Authors final response #11:**

No changes are made to the manuscript

**Reviewer #1 Comment #12**

- P.8 L-25: "at the bottom" - Figure 3: Please change the colour scale to a continuous one, otherwise it is difficult to read the map.

**Authors response #12:**

We will update the color scale as suggested for Figure 3.

**Authors final response #12:**

Colormap has been updated

**Reviewer #1 Comment #13**

- I suggest merging Figures 5 and 6.

**Authors response #13:**

**Authors final response #13:**

No changes are made to the manuscript

**Reviewer #1 Comment #14**

 - P.12 L-9: While the text indicates that December and January are not available, Figure 8 shows 11 months of data. How is that possible?

**Authors response #14:**

This is a mistake in axis settings. It is correct the December and most of January are omitted for the study due to polar darkness. We will correct the axis.

**Authors final response #14:**

Axis has been changed, months added, and ticks fixed

**Reviewer #1 Comment #15**

- P.15 L-29: This result contrasts with other studies showing a shortening of melt season and earlier onsets https://doi.org/10.1175/2010JCLI3644.1

**Authors response #15:**

It is generally true in the Northern Hemisphere that the shortening of the melt season and earlier onset of melt is observed. However, there are abnormalities for local areas. Our results suggest that this is true for Iceland and are supported by other recent similar work. (https://doi.org/10.1016/j.jag.2019.04.003)

**Authors final response #15:**

Please see added text in Author final response #1:

**Reviewer #1 Technical corrections/clarifications:**

- p.2 L-8: What is the order of the references? It is not alphabetical and not old to new.

This will be corrected

**Fixed**

- P.2 L-21: Remove extra brackets in (Fig 1,2)

This will be corrected

- P.4 L-29: I suggest changing "main objective" for "aim", since after this sentence a first and second objective are presented.

This will be corrected

**Fixed**

- P.5 L-14: Join the two references.

This will be corrected

**Fixed**

- Figure 9: Please increase figure size if possible.

This can be done in typesetting of the manuscript

This can be done in typesetting of the manuscript

Anonymous Referee #2

**Response to Anonymous Reviewer #2**

**Author response is in red**

**Final author response is in green**

**Reviewer #2 General comments:**

Cloud cover and persistence is a substantial obstacle for monitoring snow presence/absence, especially in locales with abundant clouds. The manuscript describes a methodology for a gap-filling approach to remedy the issue. The resultant product is compared against ground observations (snow absence/presence) and coincident higher-resolution imagery. The product is analyzed for landscape characteristics and temporal trends. The manuscript presents an advance in analysis of snow duration and site characteristics, especially for such a heavily cloud-dominated Iceland snow pack. Analyses such as this, performed at moderate resolution, add insight into snow variability and temporal trends that are useful and creative. As presented, the conclusions reached are carefully supported by the methods and results, and the structure of the manuscript makes it relatively easy to follow the authors' lines of thought. The paper is referenced well overall. The manuscript is hindered most by its writing, rather than its content. Substantial effort to make the writing efficient, precise, and concise throughout would go a long way to improve its clarity and make it accessible and elegant. In some places there is too much description that can be reduced. Scientifically, the paper is solid and informative. The manuscript can be improved by addressing important work done on published snow cover trends in the Northern Hemisphere and Iceland (e.g., Dietz et al.) in the Discussion.

We would like to thank Anonymous Reviewer #2 for very useful comments and a general positive feedback about our submitted manuscript.

The authors will review the manuscript carefully to make it more precise as suggested and address recent important work done by others.

**Author final response #0:**

The following text has been added:

In the Introduction section

In the Northern Hemisphere the spring snow cover extent has decreased significantly, influencing the dynamics of spring melt intensity and timing in recent years (Adam et al., 2008; Barnett et al., 2005; Choi et al., 2010; Hori et al., 2017). Various studies using remotely sensed data, observations and climate models unanimously agree that on the Northern Hemisphere scale snow cover extent and duration is receding by 2.5 to 10 days/decade depending on the study period (Eythorsson et al., 2019; Fontrodona Bach et al., 2018; Choi et al., 2010; Hori et al., 2017; Wang et al., 2018; Liston and Hiemstra, 2011). On regional scales snow cover changes can vary depending on local climatology and its variability. Future

projections with warming trends product less precipitation to fall as snow and snow melt to occur earlier in spring, affecting runoff and water resources downstream (Vaughan et al., 2013; IPCC, 2013).

In the Conclusion section:

These results are consistent with previous findings that suggest that an slight increase in snow cover extent/area is observed in Iceland (Eythorsson et al., 2019;Wang et al., 2018) even though a general decreasing snow cover extent/area and shortening of the melt season in the Northern Hemisphere is reported in many other studies (Choi et al., 2010; Hori et al., 2017).

Another influencing factor for onset of melt and melt enhancement is radiative forcing by light-absorbing particles (Painter et al., 2018; Skiles et al., 2018). Due to frequent volcanic eruptions in Iceland volcanic ash and tephra can be transported great distances (Gudmundsson et al., 2012; Július Sólnes et al., 2013). The volcanic eruptions in Eyjafjallajökull in 2010 and Grímsvötn in 2011 took place in April and May, respectively, and are within the MODIS data period (2000 - 2018). Figure 9 and 12 show no clear sign of melt enhancement for spring 2010 and 2011 although albedo and summer melt of Icelandic

**Specific Comments**

**Reviewer #2 Comment #1**

Pg. 1, line 20-21. ". . . and low thermal conductivity which is dominating for the growing season length of vegetation and plants (Keller et al., 2005)." How is snow's thermal conductivity related to growing season length, let alone a dominant for determining growing season length? Thermal conductivity is indeed important for flora, fauna, and soils in winter; but what ties are there between thermal conductivity of snowpacks and growing season length? In addition, why "vegetation and plants" in the sentence? Isn't vegetation comprised of plants?

**Author response #1:**

This is poorly worded. The authors suggest changing the text to:

"... and isolating properties which can influence the length of the growing season."

**Author final response #1:**

Changed to:

… and isolating properties which can influence the length of the growing season

**Reviewer #2 Comment #2**

The description of Icelandic land cover is especially relevant (Pg. 2, lines 12-14). The sparse and bare can be envisioned, but what is meant by "semi-natural" vegetation?

**Author response #2:**

This is poorly worded and translated wrong from Icelandic. It should say non-vegetated land classified at areas where vegetation cover is less than 10%

We will change the sentence accordingly.

**Author final response #2:**

Change to:

About 50 % of Icelands land area is classified as open spaces and bare soils with sparse vegetation and 37 % as non-vegetated where vegetation cover is less than 10 %, these two types include most of the central highlands.

**Reviewer #2 Comment #3**

Figure 2. By the time the reader arrives at this Figure, there has been no introduction of what size pixel is being referenced. Perhaps hectares or sq. km would be more useful as a y-axis variable.

**Author response #3:**

We will change this to square kilometers

**Author final response #2:**

Changed to square kilometers

**Reviewer #2 Comment #4**

Pg. 3, lines 9-11 is confusing: "A system of reservoirs and diversions store melt water during the spring freshet which generally consists of a seasonal snow melt period (April - June), a glacier melt period (June - September) and precipitation in the fall (August - October)." If you replace "spring freshet" with "year," it makes sense, but I'm not sure if this captures the intended point.

**Author response #4:**

The following re-write is suggested for clarification:

"A system of reservoirs and diversions store melt water during melt season in the spring and summer which generally consists of a seasonal snow melt period (April - June) followed by a glacier melt period (June - September). As glacier melt recedes in the fall liquid precipitation is a large contributor to inflow (August - October)."

**Author final response #4:**

"A system of reservoirs and diversions store melt water during melt season in the spring and summer which generally consists of a seasonal snow melt period (April - June) followed by a glacier melt period (June - September). As glacier melt recedes in the fall liquid precipitation is a large contributor to inflow (August - October)."

Pg. 5, lines 23-26. At times there is too much detail in the manuscript, and this is a good example of text that can be cut. "The data were downloaded from the United C2 States Geological Survey (USGS) (https://earthexplorer.usgs.gov/) using bulk download utilities. Landsat scenes that cover Iceland are numbered from 224-13 to 216-13, 224-14 to 215-14, 223-15 to 215-15 and 219-16 to 216-16 in the Worldwide Reference System 2 (WRS2), a total of 32 Landsat footprints (USGS, 2018)." Just referencing the source of the data and website in the previous sentences should suffice.

**Author response #5:**

The authors agree that this is too much detail. We will remove the above text as suggested. Also, similar details will be removed in the manuscript.

**Author final response #5:**

Some technical details in Section 2 (Data) has been removed as suggested

**Reviewer #2 Comment #6**

Organization. In Section 2 (Data), the ground observations are described first. In Section 3 (Methods), the Landsat/Sentinel data are described first. Perhaps 3.2 in-situ data processing, could be moved to 3.1 to maintain that structure? It goes back to ground observations first in Results (4.1.1)

**Author response #6:**

This has been suggested by reviewer 1 as well and we will change this

**Author final response #6:**

Order has been updated in Data and Methods. In situ, Modis and Landsat/Sentinel

Pg. 6, lines 22-25. This part discusses resampling for sentinel data, but there is no corresponding parallel description of this for Landsat data in the first paragraph of Section 3.1. More important, what sort of resampling was used to shift the pixels from 30 and 20 m resolution to 500 m?

**Author response #7:**

We suggest the following edits in Pg.6 L22-25:

Data were processed at 20 m and 30 m spatial resolution for Sentinel 2, and Landsat 7 and 8, respectively. Data were then resampled to the Modis data grid at 500 m spatial resolution using GDAL utilities (reference) with an average resampling method.

**Author final response #7:**

Added to the text:

Data were then resampled to the Modis data grid at 500 m spatial resolution using GDAL utilities (GDAL,2019) with an average resampling method.

**Reviewer #2 Comment #8**

What impacts are expected from the scale disparity going from 30/20 to 500 m?

**Author response #8:**

This is discussed in Pg. 9, lines 11-12: "The screening reveals that disagreement was mainly located at snow cover boundaries, i.e. where snow free land meets snow covered land as well as boundaries of clouds and land". These are the main effect we observe during the resampling of the data.

**Author final response #8:**

No change in manuscript

Of the pixels resampled, how much snow-covered classifications went to snow-covered areas or vice-versa? To elucidate, does snow cover largely disappear at once, or do lingering drift areas remain?

**Author response #9:**

The GDAL average resampling method converts the higher resolution satellite data to snow if more than 50% of merged pixels are classified as snow within the MODIS pixel area. Lingering drift areas need to compromise more than 50% of the Modis pixel to be classified as snow.

**Author final response #9:**

No change in manuscript

**Reviewer #2 Comment #10**

What sensitivity is in the snow-classification developed from MODIS to how much pixel area is snow covered before that threshold of snow presence/absence is crossed for Iceland?

**Author response #10:**

This is based on the NDSI index. Various NDSI Snow Cover Quality test are applied during the calculation of snow cover and supplied with the MOD10A1 granule. The sensitivity has not been investigated here.

**Author final response #10:**

No change in manuscript

**Reviewer #2 Comment #11**

Some discussion of these scale issues and inherent differences would be appreciated. Later on in the manuscript, the line (Pg. 9, lines 11-12), "The screening reveals that disagreement was mainly located at snow cover boundaries, i.e. where snow free land meets snow covered land as well as boundaries of clouds and land," is intriguing. It seems like more should be said about these boundaries and what is and isn't captured in the approach and validation with higher-resolution data.

**Author response #11:**

Higher resolution data captures indeed more details while MODIS would see a more mixed pixel. Spectral unmixing for example could prove benefits to this problem but as the data is mostly used for validation purposes it is not further pursued.

**Author final response #11:**

No change in manuscript

Pg. 8, lines 3-5. "To classify the remaining unclassified pixels information about location (Latitude, Longitude), elevation and aspect to account for earlier melting of south facing slopes are derived to apply for a gap filling algorithm." This is unclear.

**Author response #12:**

Suggested re-write to clarify:

"To classify the remaining unclassified pixels information about pixel location (Latitude, Longitude), pixel elevation and pixel aspect are derived to use for the gap filling algorithm."

**Author final response #12:**

Changed to:

To classify the remaining unclassified pixels information about pixel location (Latitude, Longitude), pixel elevation and pixel aspect are derived to use for the gap filling algorithm.

**Reviewer #2 Comment #13**

Figure 3. The color bars for the ratio of agreement aren't intuitive. Orange and Red bracket purple, blue, and green. The error magnitude would be better understood if the color scheme made a more natural color progression from high (warm/cold) to low (cold/warm).

**Author response #13:**

We will change this

**Authors final response #13:**

Colormap has been updated

Pg. 9, lines 3-4. "Pixel density range from 110, 30 and 90 for Landsat 7, Landsat 8 and Sentinel 2, respectively." This sentence is unclear.

**Author response #14:**

We suggest the following re-write for clarification:

Pixel density, i.e. number of overlappping pixel for the study period, range from 110, 30 and 90 for Landsat 7, Landsat 8 and Sentinel 2, respectively."

**Author final response #14:**

Changed to:

Pixel density, i.e. number of overlappping pixel for the study period, range from 110, 30 and 90 for Landsat 7, Landsat 8 and Sentinel 2, respectively.

**Reviewer #2 Comment #15**

Pg. 9, lines 5-6. "Visually the agreement is good in all cases with R2 values 96 %, 92 % and 95 % for Landsat 7, 8 and Sentinel 2 respectively." This statement isn't in agreement with Figure 4B, where the R2 is listed as 0.72 and the MSE seems high.

**Author response #15:**

This is a typo. 0.72 in figure 4B should read 0.92

**Author final response #15:**

Figure has been fixed

**Reviewer #2 Comment #16**

Pg. 11, lines 16. "After applying a temporal aggregation to the data unclassified pixels still remained in the dataset." Please tell us more about that here; how many? What percentage?

**Author response #16:**

As mentioned P11 L9 the improvement to pixel temporal aggregation ranges from 70% down to 14% depending on the number days selected to aggregate. This is shown in Figure 7 which is refeered to in P11 L1.

**Author final response #16:**

No change in manuscript

Figure 8. Wait, on page 12 line 9 we learned Nov-January data were not there due to darkness, and the Figure presents 12 months in a year on the X-Axis. Day of year on the X-Axis should be in DOY, not months. On the Y-Axis, why not use one tick per year instead of 0.5 year?

**Author response #17:**

This is an error at our end in the axis extent. We will change axis and change the minor tick markings.

**Authors final response #17:**

Axis has been changed, months added and ticks fixed

**Reviewer #2 Comment #18**

It would be more useful if this work were placed in a similar context with published analyses of snow cover trends. There's no discussion on volcanic impacts on snow duration. There are no contrasts provided with other published results/trends for snow cover, even at Northern Hemisphere scales. Claiming that increased glacial mass balance in Iceland is interesting, but may not be identical to what is being observed/measured in this project.

**Author response #18:**

The authors will review the manuscript carefully to make it more precise as suggested and address recent important work done by others. Impacts of volcanic eruptions will be mentioned.

**Author final response #18:**

Please see Author final response #0:

**Reviewer #2 Comment #19**

Figure 10 is interesting. It would be helpful to add a small black line to separate the Feb-Nov full dataset analyses from the bi-monthly comparisons comprising the top.

**Author response #19:**

This will be added

**Authors final response #19:**

of figures. This would results in an smaller figure on the page for the viewer where as due to detail in the maps we would like to keep the size as large as possible.

**Reviewer #2 Technical Corrections**

The paper could be shortened a bit with increased efficiency.

The authors will review the manuscript carefully to make it more precise.

"Modis" should be "MODIS" throughout.

We will change to MODIS

Has been changed to MODIS

The manuscript has a comma shortage, and there are a number of single-sentence redundancies throughout where identical words are used repeatedly in the same sentence or adjacent sentences.

The authors will review the manuscript carefully to reduce redundancies.

Pg. 2 Line 20, "higher altitudes" could be "high-altitude"

We will change this

Higher altitude glaciers is removed. The sentence is:

In the highlands this leads to the formation of a seasonal snow pack and the sustainment of glaciers

Figure 1. The green markers are hard to see on the dark gray background.

Maps has been updated and made more readable

We will change this

Need a "growing" between "vegetation" and "season" (Pg. 13, line 2)

We will change this

Changed